# Advanced Feature Extraction Methods from Images of Drillings in Melamine Faced Chipboard for Automatic Diagnosis of Drill Wear

**DOI:** 10.3390/s23031109

**Published:** 2023-01-18

**Authors:** Izabella Antoniuk, Jarosław Kurek, Artur Krupa, Grzegorz Wieczorek, Michał Bukowski, Michał Kruk, Albina Jegorowa

**Affiliations:** 1Institute of Information Technology, Warsaw University of Life Sciences, 02-776 Warsaw, Poland; 2Institute of Wood Sciences and Furniture, Warsaw University of Life Sciences, 02-787 Warszawa, Poland

**Keywords:** optical tool state recognition, artificial intelligence, drill wear classification

## Abstract

In this paper, a novel approach to evaluation of feature extraction methodologies is presented. In the case of machine learning algorithms, extracting and using the most efficient features is one of the key problems that can significantly influence overall performance. It is especially the case with parameter-heavy problems, such as tool condition monitoring. In the presented case, images of drilled holes are considered, where state of the edge and the overall size of imperfections have high influence on product quality. Finding and using a set of features that accurately describes the differences between the edge that is acceptable or too damaged is not always straightforward. The presented approach focuses on detailed evaluation of various feature extraction approaches. Each chosen method produced a set of features, which was then used to train a selected set of classifiers. Five initial feature sets were obtained, and additional ones were derived from them. Different voting methods were used for ensemble approaches. In total, 38 versions of the classifiers were created and evaluated. Best accuracy was obtained by the ensemble approach based on Weighted Voting methodology. A significant difference was shown between different feature extraction methods, with a total difference of 11.14% between the worst and best feature set, as well as a further 0.2% improvement achieved by using the best voting approach.

## 1. Introduction

Machine learning (ML) algorithms grow increasingly popular in various, industrial applications. Using the right ML method in different processes with varying complexity can significantly improve overall performance and solve key problems. At the same time, finding the right set of data to appropriately train selected algorithms can pose some issues. One of the key elements in this process is choosing the appropriate method to extract features to be used.

The wood industry, in general, and furniture manufacturing, specifically, is one of many areas in which the nature of the production environment poses a series of problems. It is especially the case with the tool condition monitoring aspect of the industry [1], where both identification of the best signals, and reducing total downtime are important factors. The most commonly used material, according to various researchers, is laminated chipboard panels [2,3,4,5]. While it is relatively cheap, this wood product also has some serious issues, apparent in the production process in the form of various wear signs and defects [3,4,5,6,7]. Additionally, its structure is difficult to predict. Depending on glue density and occurrence of air pockets, it can contain harder and softer spaces. In the case of the drilling process used with such material, pinpointing the exact moment when the drill needs to be exchanged is an important problem, with significant need for automation [8,9,10].

The topic of tool condition monitoring in the case of wood based industries, including furniture manufacturing, is not a new one. There are numerous solutions already available [11,12,13,14,15,16,17,18,19]. A significant amount of them use large arrays of sensors, measuring parameters such as acoustic emission, noise, vibrations, cutting torque, feed force, or other, similar ones [20,21,22,23,24]. There are also quite a few approaches based on ML algorithms, ranging from ones focusing mainly on monitoring the overall machining process [25,26], to approaches meant to recognize various species of wood. One of the key advantages of ML algorithms is that they can be adjusted to deal with various problems [27]. At the same time, all such solutions require a well-matched set of features to be incorporated in the training process.

Another interesting set of approaches moves away from large sets of sensors in favor of using simplified input in the form of images. Especially, transfer and deep learning methodologies were proven to be accurate in such applications [28,29]. Additional improvements were made with data augmentation and classifier ensemble approaches [30,31]. The type of used classifiers also significantly influenced the overall solution quality [32,33,34,35].

One common element for all the ML-based solutions is the need to extract the set of features to be used in the algorithm training. Drill wear recognition is a process, where various possible parameters can be taken into account. It is especially the case when images of drilled holes are considered, since apart from easily calculated parameters, other ones can be derived from image structure. There is a need for a solidified and systematic approach for choosing the best feature extraction methodology, pointing out potential gains that different methods can provide.

In this work, an evaluation of various feature extraction methodologies is presented. The problem used as a base for evaluation is a well-defined one. Various works present working solutions with good accuracy [8,9,20,24,28,29,30,31,33,34,35,36,37,38], including both image- and sensor-based approaches. Similarly, as in most of them, three wear classes were defined to describe the drill bit state. A total of five classifiers were trained using a feature set obtained with different extraction methodologies. The influence of those methods over the model quality was then evaluated. Additional tests focused on checking the influence that the chosen voting approach introduces to ensemble classifiers. Each instance of the classifier was tested in terms of overall accuracy, general solution parameters and number of critical errors performed (mistakes between border classes).

The rest of this work includes the detailed description of the used data set and methods in Section 2, presentation of obtained results and related discussion in Section 3 as well as overall conclusions in Section 4.

## 2. Materials and Methods

### 2.1. Data Collection

The images for the experiments were collected in an environment similar to the production one. During the drilling process, an automated computerized numerical control (CNC) workstation was used (Busselato Jet 100 model, Piovenne Rochette, Italy). Drilling was carried out with standard FABA drill (12 [mm] diameter, model WP-01, Figure 1), with rotation speed of 4500 RPM and feed rate of 1.35 [m/min]. Those parameters were chosen in accordance with the manufacturer’s guidelines. Wood material was laminated chipboard (KRONOPOL, model U 511, thickness of 18 [mm]), which is a standard in the furniture industry.

A total of 610 profiles were obtained from the laminated chipboard panel (Figure 2). Dimensions of each profile were 300 × 35 × 18 [mm], containing 14 holes in the central part, with equal distances. This ensured minimal influence of material tensions on the hole quality, and overall homogeneous work area. In the entire chipboard panel, a total of 8540 holes were prepared. In order to evaluate the drill state, it was regularly checked using the Mitutoyo microscope (model TM-500, Kawasaki, Japan). The drill could be assigned to one of three wear classes:Good (Green)—a new and good to use drill;Worn (Yellow)—drill in a warning state, still good enough, but might require replacement soon;Requiring replacement (Red)—unusable drill, that should be replaced immediately.

During the checkups, the drill state was classified using wear parameter (W), denoting the difference between width of the cutting edge for the brand-new drill (checked near the outer corner) and width of the same space for the evaluated tool. This parameter is measured in [mm] and was divided into following ranges, according to manufacturer’s specification:Good: W < 0.2 [mm];Worn: 0.2 [mm] < W < 0.35 [mm];Requiring replacement 0.35 [mm] < W.

Obtained profiles were then scanned for further operations. Image quality was set at 1200 [dpi]. For sample profiles containing holes drilled by tool with different wear quality see Figure 2.

### 2.2. Image Preparation

After scans of 610 chipboard profiles were obtained, images were divided into separate files for each of the holes. Storing them in such a way allowed easier analysis of individual cases as well as automation of the further image processing. Full structure of the used data set is presented in Table 1.

While most feature extraction methodologies worked on original images, hand-crafted features required additional image processing operations. For that purpose, segmentation was performed (Figure 3), with the following operations:Loading original hole image file;Contrast enhancementBlack and white image conversion with fixed threshold value (”bw”)Black and white image conversion with adaptive threshold value (”bw2”)Hole threshold sum (“bw”+“bw2”)Hole fillingLabellingArtefacts removalSaving the resulting image.

A contrast enhancement operation was used to highlight the edge detail and remove the gradient value resulting from uneven lighting. Final values were calculated according to Equation (Equation 1):(1)$$f\left(x\right)=\left\{\begin{array}{cc} 0, & \mathrm{value}\;\mathrm{lower}\;\mathrm{than}\;0.5\times 255.\\ 255, & \mathrm{value}\;\mathrm{higher}\;\mathrm{than}\;0.9\times 255. \end{array}\right.$$

Further operations involved two-stage image conversion to black and white values, to accordingly point out the actual edge of the hole. The first of those methods involved standard conversion with the set threshold equal to 0.1. The second threshold was the basis for unifying the hole using the adaptive method. As a result, the exact outline of the hole was obtained (excluding center), without losing any data about edges or imperfections related to the laminate damage.

The result of this process was a set of clean images, accurately outlining the actual hole shape for each example. The images were then stored and used in further operations.

### 2.3. Features Extraction Methods

In order to evaluate the influence of different features on the overall accuracy of the classification, a set of approaches was chosen and tested. As a result, five initial sets of features were prepared. The total number of features obtained for each of the chosen extraction methods is outlined in Table 2.

#### 2.3.1. Wavelet Image Scattering Decomposition Using Complex-Valued 2D Morlet Wavelets

Wavelet scattering, also called “scattering transform” or “wavelet transform”, uses the convolution of input signals based on object similarity. The confluence of two similar values allows for easy detection of local dependencies (correlations).

The historical context of wavelet scattering begins with the history of the Fourier transform, a fundamental signal-processing technique. The disadvantage of the Fourier representation is its instability in signaling deformations at high frequencies. This instability is due to the sine wave’s inability to locate frequency information [39].

The wavelet transform solves this problem by decomposing the signal into a family of wavelets with different dilations. The resulting wavelet representation locates the high-frequency components of the signal. Since the wavelet operator itself does the translation, the resulting representation becomes covariant in it—shifting the signal also shifts its wavelet coefficients. This makes it difficult to compare translated signals. Translation invariance is key to tasks such as classification.

The wavelet transform is based on the dot product operation, i.e., the kernel. The core is a function (called a wavelet) that satisfies the requirements for time-frequency analysis. The scale parameter *a* changes the spectrum of the wavelet in the frequency domain, and the offset parameter *b* changes the spectrum of the wavelet in the time domain.

The decomposition function used was based on a Morlet wavelet with real and complex terms plotted in the time domain [40,41]:(2)$$\Psi_{\sigma }\left(t\right)=c_{\sigma }\pi^{-\frac{1}{4}}e^{-\frac{1}{2}t^{2}}\left(e^{i\sigma t}-\kappa_{\sigma }\right)$$
where $\kappa_{\sigma }$ is admissibility criterion:(3)$$\kappa_{\sigma }=e^{-\frac{1}{2}\sigma^{2}}$$
and $c_{\sigma }$ is the normalization constant:(4)$$c_{\sigma }={\left(1+e^{-\sigma^{2}}-2e^{-\frac{3}{4}\sigma^{2}}\right)}^{-\frac{1}{2}}$$

Accordingly, the fast Fourier transform is equal to:(5)$${\hat{\Psi }}_{\sigma }\left(\omega \right)=c_{\sigma }\pi^{-\frac{1}{4}}\left(e^{-\frac{1}{2}{\left(\sigma -\omega \right)}^{2}}-\kappa_{\sigma }e^{-\frac{1}{2}\omega^{2}}\right)$$

It is a Gaussian window sine wave that performs the convolution of indexed wavelets ($\Psi_{\nu }$) at different frequency locations, and the wavelet transform ($\Psi_{x}$) is a set of spreading coefficients.

For the used data set, this method produced total of 53 separate features.

#### 2.3.2. Deep Embedding Features from Pretrained Deep Learning Network

Convolutional networks are hierarchical, multi-layered neural networks. The name historically derives from the functional model proposed by Hubel and Wiesel in 1959, after a series of studies conducted on the reaction of cat neurons. The development of practical research in the form of inspiration for digital implementation was the presentation of the “neocognitron” [42]. The concept was used to recognize Japanese words and analyze text using learning networks.

Like wavelet scattering, convolutional networks are based on the mathematical operation of convolution. The common feature of all neural networks is that their structure consists of “neurons” connected by “synapses”. Synapses are associated with weights, i.e., numerical values, the interpretation of which depends on the given model. Each layer is connected to the next and previous ones. The result of the layer analysis is directed to the next one. It also takes into account the subsequent refinement data, which have significant characteristics affecting the analysis of the corresponding results.

The network used was ResNet-18. It is a convolutional neural network with a depth of 71 layers but the number 18 refers to the core (deep) layers the architecture is based on: the convolution and fully-connected layers [43]. The purpose of this network was to enable the functioning of a large number of convolutional layers. Excess layers cause a loss of key functionality, i.e., the quality of the output data. Due to the presence of numerous layers, multiple downward multiplications of each layer causes the gradient to decrease and thus “disappear”, leading to network saturation or performance loss.

The network is primarily designed to classify images into more than 1000 categories, thanks to which the pre-trained one can assign almost flawlessly the definitions of input data of many objects. The ResNet-18 architecture is 71 layers (70 + 1 as the target “output”) containing 18 deep layers (convolution and fully-connected layers).

The method used for feature extraction took features from layer 68. This layer was chosen since it contains all embedded features from images that help in classification. In this case, 512 features were generated for each image. The network is pre-trained on a very large set from the *ImageNet* database, so it is not necessary to train it on new data to extract the features.

#### 2.3.3. Shallow Embedding Features from Pretrained Deep Learning Network ResNet-18

In order to evaluate which type of features adapt better to the presented problem, another set was extracted from the ResNet-18 35th layer. The approach is the same as with the features obtained from the 68th layer. In general, earlier layers extract fewer, shallower features, have higher spatial resolution, and a larger total number of activations. It is the last layer to transmit 128 new features. This ensures optimal spatial distribution at the level of 28 × 28.

#### 2.3.4. Hand Crafted Features

In the feature extraction process, physical aspects were also taken into account, based on the analysis of photos of the holes. When it comes to furniture manufacturing, types of imperfections can have a different impact on the final product quality. In that regard, a hole with a single, long chip is far worse than one with many, short chips, situated close to the edge. In the second case, the imperfections can be hidden during furniture assembly. In the first case, the final product needs to be discarded. A set of hand crafted features was prepared, to represent those properties, which contained total of nine characteristics:Radius of the smallest circumscribed circle of the hole (Figure 4a)Radius of the largest circle inscribed in the hole (Figure 4b)Difference of hole radiiArea of holesConvex surface areaCircumferenceThe major axis of the ellipse described in the imageMinor axis of the ellipse described in the imageMassiveness (surface area/convex area)

After incorporating the initial assumptions about the main focus of the classification process, the features were then evaluated using the classical form of Fisher’s measure. The diagnostic value of the k-th feature in recognition of two classes (*i*-th and *j*-th) is defined by Equation (Equation 6):(6)$$W_{k}\left(i,j\right)=|\frac{\mu_{k}\left(j\right)-\mu_{k}\left(i\right)}{\sigma_{k}\left(j\right)+\sigma_{k}\left(i\right)}|$$

This analysis produced a sorted value list for the used features in the following order: 5, 4, 6, 9, 3, 7, 2, 8, 1, with decreasing importance.

#### 2.3.5. Extracted Histogram of Oriented Gradients (HOG) Features

The histogram of oriented gradients (HOG) is a so-called feature descriptor, such as SIFT (scale invariant feature transform) or SURF (speeded-up robust feature) [44,45]. It is used for feature extraction in computer vision or computer vision systems. HOG focuses on the structure or shape of an object. With edge detection, the only identification checks if a pixel is an edge or not. HOG additionally defines an important element from this point of view–direction. This is possible thanks to the gradient and the orientation of the edges.

The image is divided into smaller areas in the analysis process. Gradients and orientation are calculated for each region. The result is the histogram mentioned in the name of the descriptor for each of these regions separately. Histograms are created using gradients and pixel value orientations.

The analysis consists of three stages—data preparation, determination of gradients and determination of orientation based on information from the gradient. Each change is determined in both directions of the 2D object—on the X-axis and the Y-axis. The differences between them are the basis of the classic Pythagorean approach to determining the extent of the gradient. In addition, the angle ϕ determining the direction is determined using these relationships.

Using HOG allowed obtaining 1296 individual features from smaller areas with single cell size equal to 32 × 32.

### 2.4. Classifiers

In the current approach, the main focus was put on the influence each feature extraction method has on classification accuracy. Therefore, a set of state-of-the-art classifiers was trained, using different feature sets to evaluate the influence each of them has on a given solution.

#### 2.4.1. K-Nearest Neighbours

The first of the chosen classifiers is K-NN—the non-parametric method and a good base for further comparison. This algorithm assigns the class of an object by calculating to which one most of its neighbours it belongs. If for the selected object more than one class has an identical number of neighbours, the distance to each of them will be checked. Final classification will be then assigned according to the neighbourhood, with smaller “distance” to the current example [30,46,47,48].

A basic version of the algorithm with Euclidean distance is not often used. In order to improve the classification accuracy, neighborhood components analysis (NCA) is usually incorporated. This algorithm maximizes a stochastic variant of the leave-one-out k-nearest neighbors score on the training set, resulting in better classification.

In regards to overall parameters used in the K-nearest neighbours method, the presented approach had the following values:K = 5metrics = ‘minkowski’leaf_size = 30

#### 2.4.2. Extreme Gradient Boosting

The second used classifier is an improved version of the classic gradient-boosting-based solution called extreme gradient boosting or XGBoost [49,50,51,52].

This method was chosen since it presents many advantages in comparison to its predecessor. It introduces regularization rules to the training and, additionally, incorporates parallel processing. In case of missing data values, it has an in-built feature to handle such elements. The same applies for the cross validation technique and tree pruning feature. Overall, the method has potential to achieves significantly better results [53].

In the presented calculations, extreme gradient boosting had the following parameters:M = 100loss function = ‘log-loss’max_depth = 3learning_rate = 0.1min_samples_leaf = 1

#### 2.4.3. Random Forest

The third selected method was Random Forest, as it incorporates the ensemble learning approach. The general idea with such algorithms is the combination of predictions performed by several base classifiers. The goal is to improve robustness in comparison to single estimator [54].

In the case of Random Forest, the estimators are decision trees. Each one of them is built using a sample from the training set, selected with a replacement and creating a random subset of samples. Moreover, a random subset of features is used for node splitting and finding the best possible split [55,56].

When individual decision trees are considered, they tend to have high variance and are prone to overfitting. The randomness element in the Random Forest approach decreases the variance and reduces the chances of such problems occurring. The injected randomness decouples prediction errors and, by averaging, some of them can cancel out. The variance reduction often leads to an overall better model.

The presented Random Forest implementation used the following parameters:min_samples_leaf = 1loss function = Ginibase classifier = Decision Tree

#### 2.4.4. Light Gradient Boosting

LGBM uses the gradient boosting method, but unlike algorithms based on random trees (for example the previously mentioned XGBoost), does not rely on sorting while finding the best split point. Instead, it is based on decision trees using the decision histogram. This provides the possibility to follow the path of the expected least loss in time [57,58].

LGBM has vertical growth (leaf-wise) that results in more loss reduction and it tends to achieve higher accuracy while XGBoost has horizontal growth (level-wise).

Light gradient boosting used the following parameters:M = 100loss function = ‘log-loss’learning_rate = 0.1reg_alpha = 0reg_lambda = 0boosting_type = ‘gbdt’

#### 2.4.5. Support Vector Machine

SVM is the classification method based on correctly mapping data to multidimensional space and applying a function for data separation in order to declare classes [59,60]. Either a hyperplane or set of hyperplanes in a high dimensional space are created based on kernel functions. The goal is to maximize the separation margin largest distance to the nearest training data points of any class (called support vectors) [61].

In the case of multi class classification, a “one-versus-one” approach is often applied, which means that m∗(m−1)/2 classifiers are constructed where m is the number of classes.

In presented calculations SVM had the following parameters:C = 100kernel = ‘RBF’gamma = 1/561

#### 2.4.6. Principal Component Analysis (PCA)

Since in the initial experiments a large overall set of features was obtained, the general assumption was that it is possible to derive the most important features from that set, significantly reducing their number. PCA is a method used to reduce the dimensionality of a data set by transforming the initial, large set of variables to a smaller one, the main cost being decrease in accuracy. With the appropriate components derived, the information loss will be minimal, with significant decrease in the number of features.

In order to use PCA, the data needs to be standardized. It is usually carried out by recalculating all values in data set to comparable scales, to avoid data with higher value ranges biasing the results. After that, the covariance matrix is calculated in order to pinpoint the possible relationships between different variables. Principal components are then determined by computing the eigenvectors and eigenvalues of the covariance matrix. Those are the new variables, constructed from linear combinations of initial variables. New values are prepared in such a way that they are not correlated, and the first component will contain the most information from the initial set. It is then a matter of how many components need to be included into the final features set, to represent an acceptable percentage of information from the initial data set.

#### 2.4.7. Voting Classifier

The idea behind the voting classifier approach is to combine conceptually different machine learning classifiers. Either a majority vote (hard vote) or the average predicted probabilities (soft vote) is then used to predict the class labels. Such a classifier can be useful for a set of equally well performing models in order to balance out their individual weaknesses [62].

In contrast to majority voting (hard voting), soft voting returns the class label as argmax of the sum of predicted probabilities. Specific weights can be assigned to each classifier via the weights parameter. When weights are provided, the predicted class probabilities for each classifier are collected, multiplied by the classifier weight, and averaged. The final class label is then derived from the class label with the highest average probability [62].

An additional approach was derived by incorporating the overall effectiveness of used classifiers. The overall weight for each classifier was calculated based on the accuracy of the results from experiments performed using initial feature sets (Table 3). The assigned values range from 1 (meaning that the classifier achieved the worst results for the current feature set) to 5 (for the best classifier). Final weighted importance was calculated as the share of the results obtained by a given classifier to the total number of points.

#### 2.4.8. General Classification Approach

For the final classification, the chosen model was trained using different features sets, both the five individual sets, as well as combined feature sets, derived during the performed experiments. Training was carried out using five-fold cross-validation. This means that each data set was treated as a test set, with the remaining four used for training, repeating this process for each set. This approach was based on an expert suggestion, and the main goal here was to minimize the risk of overfitting the model. In general, such an approach will result in a more confident model. Not only is the entire data set is used, but the risk of some accidental bias occurring due to its structure is minimized. At the same time, the model will better “understand” used examples and data set structure.

Overall, a total of 38 versions of the classification methodology were prepared:twenty-five with initial features—each classifier was trained using all initial feature sets;five classifiers, based on all features obtained through all feature extraction methods;five classifiers based on set of features obtained by using the PCA (principal component analysis) method on the set containing all features;hard voting approach based on five classifiers, trained on the whole feature set;soft voting approach based on five classifiers, trained on the whole feature set;weighted soft voting approach based on five classifiers with weights, trained on the whole feature set;

Each classifier was individually evaluated using the same set of parameters. The summary results were then represented using confusion matrix and table with detailed accuracy values.

## 3. Results and Discussion

The general approach used in this work trained classifiers on different sets of features in order to evaluate extraction methodologies, and general characteristics of feature sets obtained from them. Each model was evaluated in terms of accuracy, overall parameters and number of performed misclassifications between different classes. Table 3 contains the general outline for the performance of each individual model.

After preparing the initial feature sets and training five selected classifiers using those sets, it can be seen that the type of extraction method used can have high influence over algorithm performance. When evaluating each classifier, it can be seen that the difference between the worst and best instances of the model were quite significant. It can also be seen that not all classifiers performed the best using the same sets of features, although instances trained on all available features achieved best results for three out of five classifiers (LGBM, RF and XGBoost). When it comes to the remaining models, KNN performed best with features extracted using the deep embedding approach, while for SVM, the shallow embedding methodology returned higher results. What is more, for all algorithms, the difference between the best and worst instances of the classifiers was higher than 3%, with the largest difference occurring for the KNN algorithm (5.24%). Confusion matrices presenting detailed results for the best instance of each classifier are presented at Figure 5a–e and Figure 6a.

Since for most cases, the best results were obtained using set containing all extracted features, the following classifiers were tested using those results. The first of them was XGBoost, also used to evaluate importance of features from initial sets. Out of a total 1998 features, the XGBoost feature selection method chose 1109. Table 4 outlines the importance of the remaining features from each initial feature set. As can be seen, two of the most important sets with over 80% set share were obtained using deep embedding and HOG feature extraction approaches. The ranking of 75 of the most important features for the XGBoost algorithm is presented in Figure 7, while the confusion matrix is shown at Figure 5f. At Figure 7, it can be noticed that the main importance groups of features are deep embedding, shallow embedding and HOG. There is a significantly larger number of HOG features in comparison to the rest of the group. However, analyzing Table 4, it is clear that features having the greatest share and impact are the ones automatically extracted by the state-of-the-art deep learning algorithms.

Both in the case of classifiers using different feature sets, as well as with PCA and XGBoost where subsets of features with highest influence are chosen, the improvement in overall accuracy can be seen. Further evaluation focused on checking the influence of voting type used for classifier ensembles. Three types of voting were used in presented approach. The first, called hard voting, used a standard majority voting approach, without any regard to individual classifier performance. The second, denoted as soft voting, used highest average probability. The final approach, called weighted voting, used modified soft voting with an additional modifier. This parameter took into account how the individual classifier performed with different feature sets, and assigned points according to the place it took in that aspect (ranging from 1 to 5). The individual classifier scores and resulting weights are presented in Table 5.

All three voting approaches achieved similar results, with the highest score achieved by the weighted voting approach (Table 6). All classifiers were trained using the same features set, containing all available features. At the same time, the difference between best and worst approach reaches only 0.4%, which is not as significant as in the case of differences between feature extraction methodologies.

While accuracy is a good parameter for evaluating the overall algorithm performance, in the presented case another important factor is the types of errors made (Table 7). From the manufacturer point of view, the worst type of misclassifications are instances where Red class will be classified as Green. In those cases, denoted as critical errors, a product with notable imperfections will be created, which can result in financial loss, since such elements need to be discarded. Similarly, as with overall accuracy, in the case of the number of Red–Green errors, a classifier trained on all features achieved the best results. In the case of voting classifiers, the hard voting approach was the best performing one. As with overall accuracy, the difference here is minimal, nevertheless the best approach improves the error score by 24 Red–Green misclassifications.

Overall, the above analysis shows, that approach used for feature extraction can have significant impact on the training process. The difference between the best and worst performing instances of the classifiers reached 5.24% in the case of a single classifier trained on different feature sets. This gap was even higher when different classifiers and feature sets were considered, especially with the introduction of voting approaches. Final improvement reached 11.14%, between overall worst classifier (KNN using Wavelet feature set) and the best one from the initial set (XGBoost trained on all features). Introduction of different voting approaches further improved the accuracy, with weighted soft voting having 0.20% better accuracy.

An additional problem that needs to be considered is the trade-off between the approach complexity and achieved accuracy. In the case of results obtained by classifiers trained on different feature sets (see Table 3), the highest accuracy was obtained by solutions trained on all available features. At the same time, the difference in terms of accuracy between this approach (XGBoost) and the best performing classifier using one of the feature sets (SVM, using shallow embedding features) is only 0.54% (accuracy equal to 64.27% vs 63.73%). At the same time the difference in complexity is significant (1998 features in contrast to only 128). This is even more visible with the hand crafted feature set for the same, SVM algorithm, with accuracy equal to 63.54% (total difference of 0.73%) with only 9 features used. Depending on the problem, such drastic improvement in solution complexity with only minimal gain may not be considered as a acceptable trade-off. Those differences are also a good indication that choosing appropriate feature extraction methodology, or a subset of features altogether, is a very important problem, that needs to be seriously considered and evaluated.

The results for the critical errors performed are not as clear, with best approaches diverging from the accuracy score. Nevertheless, using different feature sets resulted in significant difference in those errors as well. In the case of Red–Green errors, the total difference reached 321 (best instance of hand crafted features which performed 1563 errors, and the same for all features with 1242 errors). Further research might be needed to exactly point out the best features for excluding chosen error types.

The main focus of the presented research was to evaluate the influence the different feature extraction methods had on overall algorithm performance. The differences between chosen sets, and further influence of voting approaches in the case of ensemble approaches is clearly present. Further research could focus either on evaluating additional feature extraction methods, and incorporating different classifiers for testing purposes.

## 4. Conclusions

In this paper, a new approach to evaluating feature extraction methodologies is presented. In the case of ML algorithms, obtaining the appropriate set of features to be used in the training process is one of the key problems, significantly influencing the final solution performance.

The presented approach used five classifiers (KNN, XGBoost, Random Forest, LGBM and SVM), trained on different feature sets. The first five feature sets were obtained using different extraction methodologies: wavelet image scattering, deep and shallow embedding features from a pretrained ResNet-18 network, hand crafted features and HOG approach. An additional feature set containing all parameters in the initial sets, and ones taken from this range using the PCA method, were added. To further evaluate the influence of the classifier, three voting approaches were incorporated. A total of 38 classifier instances were trained.

It can be clearly seen that the type of features obtained using different extraction methodologies has a significant impact on the final model accuracy. The difference between best and worst solution instances reached 11.14%. The same goes for the Red–Green misclassifications. The difference between best and worst classifiers reached a total of 321 errors (between approaches based on hand-crafted and all features).

While the accuracy score presents clear results, there are also additional factors to consider. While the best classifier (XGBoost, trained on all 1998 features) achieved an accuracy equal to 64.27%, the best solution based on a significantly smaller set of parameters (SVM using 128 features obtained using shallow embedding methodology) was less accurate only by 0.54%. Additionally, nine hand-crafted features (also using SVM) achieved 63.54% accuracy and were only 0.73% worse from the best solution in that set. When feature extraction methods are considered, the problem of trade-off between solution complexity and overall solution accuracy is also an important factor. As can be seen, the impact of appropriate extraction methodology is even more visible in that case.

In addition to algorithm performance, the influence of the used voting methodology was checked. The same set of five classifiers was tested, using three approaches: hard, soft and weighted soft voting. While the differences between those classifiers are still visible, they are far less impactful than the changes caused by using different feature sets. Additionally, a classifier with a hard voting approach performed best in terms of critical errors, while one with weighted soft voting achieved best accuracy.

Overall, it can be noted that extracted features can significantly impact the solution quality. Pointing out the best performing set is not always straightforward. Additionally, the above evaluation clearly shows that choosing the appropriate feature extraction method can be a key element influencing the overall solution quality and complexity. Depending on the type of problem and general specifications, different approaches need to be considered.

## Figures and Tables

**Figure 1 sensors-23-01109-f001:**
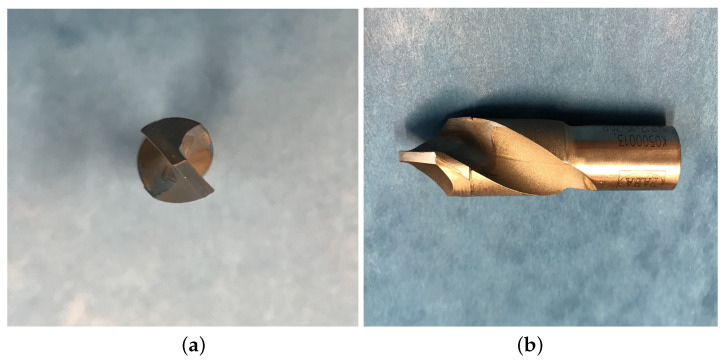
The FABA WP-01 drill used during experiments: top (**a**) and left (**b**) view.

**Figure 2 sensors-23-01109-f002:**
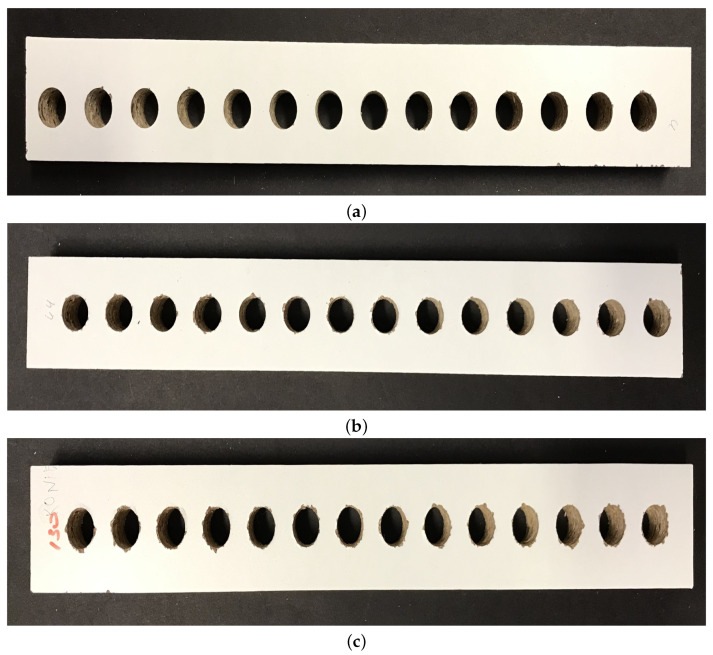
Example initial profile scans with drilled holes representing three recognized wear classes: Good (**a**), Worn (**b**) and Requiring replacement (**c**).

**Figure 3 sensors-23-01109-f003:**
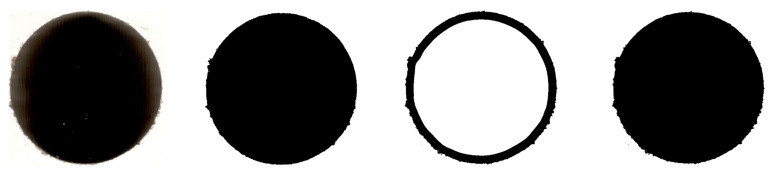
Steps of image segmentation for the hand crafted feature set (left to right): input scan, “bw”, “bw2”, “bw+bw2”.

**Figure 4 sensors-23-01109-f004:**
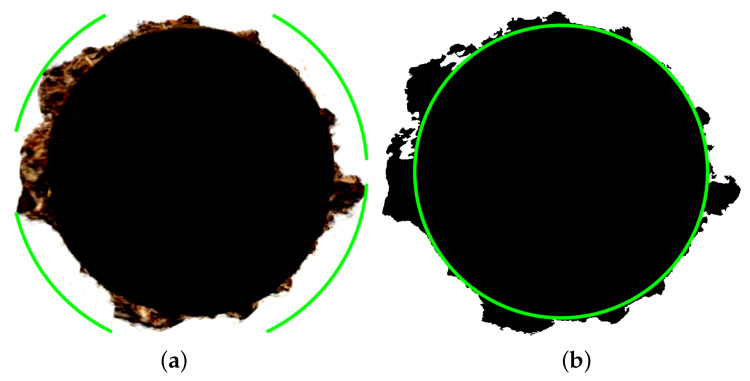
Comparison of hole radius—circumscribed (**a**) and inscribed (**b**) hole radius.

**Figure 5 sensors-23-01109-f005:**
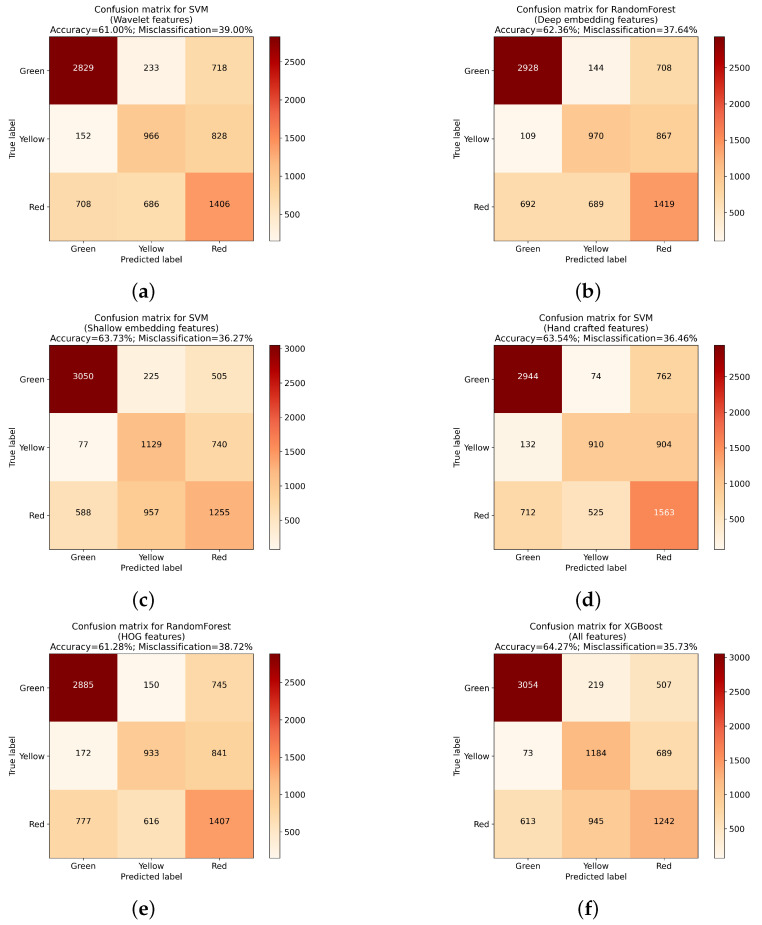
Confusion matrices for the best classifier trained using each feature set. (**a**) SVM using Wavelet transform; (**b**) Random Forest using deep embedding; (**c**) SVM using shallow embedding; (**d**) SVM using hand crafted features; (**e**) Random forest using HOG; (**f**) XGBoost on all features.

**Figure 6 sensors-23-01109-f006:**
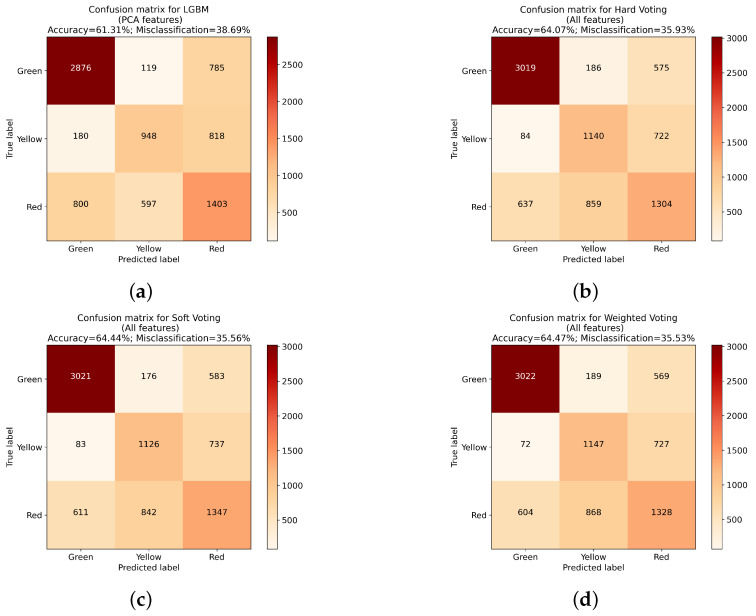
Confusion matrices for the best classifier trained using each feature set for PCA-based and voting-based approaches. (**a**) LGBM using PCA on all features. (**b**) Hard voting classifier on all features. (**c**) Soft voting classifier on all features. (**d**) Weighted soft voting approach on all features.

**Figure 7 sensors-23-01109-f007:**
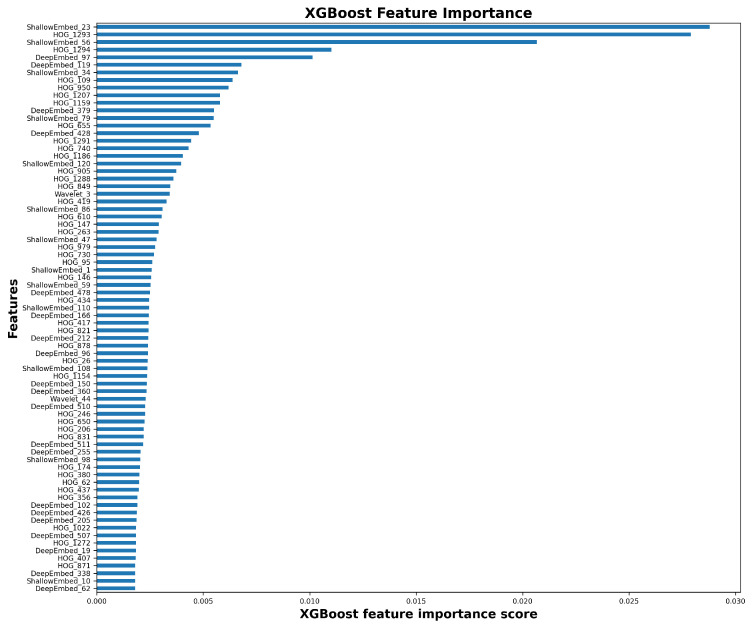
Feature importance ranking for XGBoost—first 75 most important features calculated from all folds used during training process.

**Table 1 sensors-23-01109-t001:** Structure of used data set, including number of images for each class for the sequentially used drills.

No. of Drill	Green	Yellow	Red	Total
Drill No. 1	840	420	406	1666
Drill No. 2	840	700	280	1820
Drill No. 3	700	560	420	1680
Drill No. 4	840	560	280	1680
Drill No. 5	560	560	560	1680
Total	3780	2800	1946	8526

**Table 2 sensors-23-01109-t002:** Total number of features for each initial feature extraction approach.

Set of Features	No. of Features
Wavelets	53
Deep embedding	512
Shallow embedding	128
Hand crafted	9
HOG	1296
Total	1998

**Table 3 sensors-23-01109-t003:** Summary of the accuracy for 5 classifiers trained using extracted feature sets. The best result for each classifier is underlined.

Feature Extraction Method	KNN	LGBM	RF	SVM	XGBoost
Wavelet	53.13	60.59	59.79	61.00	60.00
Deep Embedding	**58.37**	62.12	62.36	61.87	62.07
Shallow Embedding	57.58	63.29	63.03	**63.73**	63.00
Hand crafted	56.44	62.66	63.05	63.54	62.68
HOG	55.11	60.33	61.28	59.96	60.13
All features	56.45	**64.16**	**63.15**	63.38	**64.27**
PCA	56.63	61.31	61.25	61.25	60.43
Highest difference (max-min)	5.24	3.83	3.36	3.77	4,27

**Table 4 sensors-23-01109-t004:** Feature importance for the feature set that was chosen by the XGBoost algorithm. For each feature extraction method number of occurrences and % share of features qualified to the model as significant.

Feature Set	No. of Occurrences	% Share
Wavelet	52	4.69%
Deep Embedding	511	46.08%
Shallow Embedding	126	11.36%
Hand crafted	0	0%
HOG	420	37.87%
Total	1109	100%

**Table 5 sensors-23-01109-t005:** The ranking and calculated importance of classifiers taking part in classification.

Feature Extraction Method	KNN	LGBM	RF	SVM	XGBoost
Wavelet	1	4	2	5	3
Deep Embedding	1	4	5	2	3
Shallow Embedding	1	4	2	5	3
Hand crafted	1	2	4	5	3
HOG	1	4	5	2	3
All features	1	4	2	3	5
PCA	1	5	3	4	2
Total	7	27	23	26	22
Weighted importance	0.067	0.257	0.219	0.248	0.209

**Table 6 sensors-23-01109-t006:** Results obtained for the three voting approaches with different types of score calculation. Classifiers were trained using all features.

Set of Features	Hard Voting	Soft Voting	Weighted Voting
All features	64.07	64.44	**64.47**

**Table 7 sensors-23-01109-t007:** Critical classification errors achieved by best instance of classifier for each feature set and voting type.

Extraction Type/Voting	Red-Green Errors	Accuracy
Wavelet	1406	61.00%
Deep Embedding	1419	63.36%
Shallow Embedding	1255	63.73%
Hand Crafted	1563	63.54%
HOG	1407	61.28%
XGBoost on all features	**1242**	**64.27%**
PCA	1403	61.31%
Hard Vote	**1304**	64.07%
Soft Vote	1347	64.44%
Weighted Vote	1328	**64.47%**

## Data Availability

Not applicable.

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
