# Peer review of "Advanced Feature Extraction Methods from Images of Drillings in Melamine Faced Chipboard for Automatic Diagnosis of Drill Wear"

_sensors, 2023, doi:10.3390/s23031109_

Round 1

Reviewer 1 Report

see the report

Author Response

Dear Reviewer, 

 We would like to thank You for the time spent reviewing our paper and providing valuable comments. We are sure that your insightful remarks led to improvements in the current version of the paper. The authors have carefully considered the comments and strived to address and implement every one of them. We hope the manuscript after careful revisions meet your high standards. The authors welcome further constructive comments. 

Below we provide the point-by-point responses. All modifications in the manuscript have been highlighted in yellow. In case of new sections and major reviews of existing ones, only the title was highlighted. 

  1. in lines 147-148 the correct indent has to be used; 

Thank You for pointing that out, the ident has been corrected.  

  1. in lines 177-178 the ResNet architecture is described according to 18 levels, but previously you were describing it according to 18 layers: you have to correct and clarify, given that also in line 186 and 310 you mention the specific 35th layer; 

Thank You for pointing that out. It is of course an inaccuracy on our part. It is a convolutional neural network with a depth of 71 layers but the number 18 refers to the core (deep) layers the architecture is based on: the convolution and fully-connected layers. Those details were corrected in the article. 

  1. in lines 235-239 you describe the use of ensemble learning, known to consist of only a concrete finite set of alternative models: so why do you underline the use of a single classifier? You have to specify what kind of methods are usually exploited in order to explain; 

In the initial version of the paper, we have applied Boosted Decision Trees. It is treated as single classifier but built using many weak classifiers (e.g., a hundred decision trees). Boosting means combining a learning algorithm in series to achieve a strong learner from many sequentially connected weak learners. In case of gradient boosted decision trees algorithm, the weak learners are decision trees. In current version we have replaced Boosted Decision Trees with Extreme Gradient Boosting. This approach has similar nature of working from theoretical point of view but much better implementation (due to this it is called “extreme”). 

Also, due to other suggestions received, we incorporated additional classifiers to the analysis. The classifiers are described in the Classifiers subsection of Materials and Methods section. Each new subsection title is highlighted in yellow. Current approach uses 5 different classifiers, as well as 3 different voting approaches, used to better show the influence of the chosen feature sets.  

  1. in Results a discussion regarding what kind of features typify a correct assignment of labels is missing. 

In the traditional machine learning approach that has been used for the last decade, industry experts suggest machine learning specialists what features are important from the point of view of modelling a given phenomenon. Today, you can still find such an approach, and it is also used in this article as Hand-Crafted features. In the modern algorithms, including deep learning approaches which, assuming large sets of data also displace traditional machine learning. Automatic methods of extracting features can be found at different levels of accuracy / resolution. In algorithm approaches such as HOG, the so-called Artificial variables are difficult to name and define, but similarly or much better capture differences in the analysed images. The problem here is to define what a given feature applies to. It is difficult to describe accurately, when it was automatically obtained by an algorithm, e.g. of the Resnet-18 network. However, these automatic extraction methods are known to work and can classify objects much more precisely with larger datasets. They also often find features that industry experts are not even able to see with such huge datasets. Due to the fact that there are a lot of these features extracted and they are "artificial", it is only possible to describe statistically how many features and from which groups are the most important from the modelling point of view. The same goas for providing information for the correct assignment of labels. Such statistics as feature importance or the % share of variables of a given type in the model. It also provides information which extraction method is worth focusing on in the future to refine the approach to the topic. Such information was included in the article in the form of tables and described in the conclusions. 

Reviewer 2 Report

The abstract does not mention the nature of the extracted features (i.e. the type of data and the nature of the problem which is solved).

ResNet-18 is a very well-known CNN. Table 2 is not adding any value, in my opinion.

Please justify the selection of the 9 handcrafted features. Why are they representative of the situation, and what is their physical/logical connection to the drill’s condition?

Feature classifier plays an essential role. Hence, using only one classifier is considered a severe study limitation.

There is no discussion of the results. Also, the contribution of the study is not clarified across the text. From my point of view, Table 4 reveals the most significant result. The combination of the extracted features yields the best result. Hence, whilst the individual feature extractors can’t provide representative features to explain more than 64% of the population, combining features improves the accuracy. The authors take this result for granted: “As could be predicted, this approach performed best, with highest overall accuracy and lowest number of critical misclassifications.”. I disagree that this performance was expected. However, the feature-combination method yields 64.97%, whilst Shallow attains 64.05%. The improvement is minor. There is a trade-off between performance and complexity. The authors can discuss this more. What method do the authors propose for this task after observing those results?

There are plenty of similar works that utilise similar datasets and methodologies. In many of them, the authors of this study are co-authors. A short search revealed the following works:

https://www.mdpi.com/1424-8220/21/23/8077

https://link.springer.com/article/10.1007/s00226-020-01245-7

https://bioresources.cnr.ncsu.edu/resources/time-efficient-approach-to-drill-condition-monitoring-based-on-images-of-holes-drilled-in-melamine-faced-chipboard/

https://www.mdpi.com/1424-8220/20/23/6978

https://mgv.sggw.edu.pl/article/view/2302

https://journals.pan.pl/dlibra/publication/98119/edition/84562/content

there needs to be a careful examination of the current study's novelty and contribution, given those published results. The authors need to convince the readers that they are presenting something new.  

Author Response

Dear Reviewer, 

We would like to thank You for the time spent reviewing our paper and providing valuable comments. We are sure that your insightful remarks led to improvements in the current version of the paper. The authors have carefully considered the comments and strived to address and implement every one of them. We hope the manuscript after careful revisions meet your high standards. The authors welcome further constructive comments. 

Below we provide the point-by-point responses. All modifications in the manuscript have been highlighted in yellow. In case of new sections and major reviews of existing ones, only the title was highlighted. 

  1. The abstract does not mention the nature of the extracted features (i.e., the type of data and the nature of the problem which is solved). 

Thank You for pointing that out. The above information was included in the abstract.  

  1. ResNet-18 is a very well-known CNN. Table 2 is not adding any value, in my opinion. 

The ResNet-18 structure was removed. 

  1. Please justify the selection of the 9 handcrafted features. Why are they representative of the situation, and what is their physical/logical connection to the drill’s condition? 

The Hand-Crafted features were selected after discussion with the industry expert, regarding the main reasons the products are discarded. When it comes to furniture preparation, certain level of imperfections along the hole edge is acceptable, especially if it remains close to the drilling. In that regard, hole with single, long chip is far worse than one, with many, short chips, close to the edge. In the second case, the imperfections can be hidden during the furniture assembly. In the first case, the final product needs to be discarded. In that aspect the Hand-Crafted features were prepared to represent those properties.  

The description outlining this was added in the subsection describing the Hand-Crafted features.  

  1. Feature classifier plays an essential role. Hence, using only one classifier is considered a severe study limitation. 

Thank You for Your suggestion. After evaluating our research, we prepared set of different classifiers to better evaluate the feature extraction methodologies. Descriptions outlining each chosen classifier were added to the Classifier subsection in Materials and Methods section. We also evaluated three different voting methods for ensemble approach. Descriptions and results throughout the entire paper were corrected accordingly.  

  1. There is no discussion of the results. Also, the contribution of the study is not clarified across the text. From my point of view, Table 4 reveals the most significant result. The combination of the extracted features yields the best result. Hence, whilst the individual feature extractors can’t provide representative features to explain more than 64% of the population, combining features improves the accuracy. The authors take this result for granted: “As could be predicted, this approach performed best, with highest overall accuracy and lowest number of critical misclassifications.”. I disagree that this performance was expected. However, the feature-combination method yields 64.97%, whilst Shallow attains 64.05%. The improvement is minor. There is a trade-off between performance and complexity. The authors can discuss this more. What method do the authors propose for this task after observing those results? 

Thank You for pointing that out. The Results and Discussion section was rewritten, due to additional classifiers being incorporated and the discussion was extended, to better outline obtained results. We also included the discussion regarding the trade-off between solution complexity and achieved accuracy. Overall, the current approach is better in terms of critical errors, since the misclassification was shifted to the less impactful ones (64.27% accuracy and 1242 misclassifications, when compared with previous 64.97% and 1385 errors). After consulting with industry experts, it was noted that for the industry it was more important to reduce the number of critical errors than improve overall accuracy. Hence the current approach is more appropriate for potential implementation. 

  1. There are plenty of similar works that utilise similar datasets and methodologies. In many of them, the authors of this study are co-authors. A short search revealed the following works: 

https://www.mdpi.com/1424-8220/21/23/8077  

https://link.springer.com/article/10.1007/s00226-020-01245-7  

https://bioresources.cnr.ncsu.edu/resources/time-efficient-approach-to-drill-condition-monitoring-based-on-images-of-holes-drilled-in-melamine-faced-chipboard/  

https://www.mdpi.com/1424-8220/20/23/6978  

https://mgv.sggw.edu.pl/article/view/2302  

https://journals.pan.pl/dlibra/publication/98119/edition/84562/content 

  1. there needs to be a careful examination of the current study's novelty and contribution, given those published results. The authors need to convince the readers that they are presenting something new.     

It is of course true, that during previous research similar datasets and methodologies were used to similar or identical applications, but the focus of this paper is put on evaluation of different feature extraction methodologies. In authors opinion, there is far too little research regarding that topic. Previously performed test led us to believe, that selecting and using appropriate features can be very significant step to achieving acceptable classification results, hence the need for the presented research. The difference each feature set introduces to the results obtained by classifier was clear in case of single used classifiers. After introducing additional classifiers, the accuracy improvement between best and worst instances of them is even higher (11.14\%). We also checked influence of different voting methodologies in case of ensemble approaches and it led to further .2\% improvement.  

Entire Results and Discussion as well as Conclusions section was rewritten, both to better point out the contribution of the paper as well as to include new results obtained during additional tests. Thank You for Your valuable suggestions, we believe that they led to significant improvement of the paper.  

Round 2

Reviewer 2 Report

The authors addressed my comments. I recommend this paper for publication.